# Natural Occurrence of Conventional and Emerging *Fusarium* Mycotoxins in Freshly Harvested Wheat Samples in Xinjiang, China

**DOI:** 10.3390/toxins17120591

**Published:** 2025-12-10

**Authors:** Weihua Zheng, Jinyi Zhang, Yi Shi, Can He, Xiaolong Zhou, Junxi Jiang, Gang Wang, Jingbo Zhang, Jianhong Xu, Jianrong Shi, Fei Dong, Tao Sun

**Affiliations:** 1Laboratory of Quality and Safety Risk Assessment for Agro-Products of Ministry of Agriculture and Rural Affairs, Key Laboratory of Agro-Products Quality and Safety of Xinjiang, Institute of Quality Standards & Testing Technology for Agro-Products, Xinjiang Academy of Agricultural Sciences, Urumqi 830091, China; 2Institute of Food Safety and Nutrition, Jiangsu Academy of Agricultural Sciences, Nanjing 210014, China; 3College of Agronomy, Jiangxi Agricultural University, Nanchang 330045, China; 4State Key Laboratory for Managing Biotic and Chemical Threats to the Quality and Safety of Agro-Products, Institute of Agro-Product Safety and Nutrition, Zhejiang Academy of Agricultural Sciences, Hangzhou 310021, China; 5College of Plant Science and Technology, Huazhong Agricultural University, Wuhan 430070, China; jingbozhang@mail.hzau.edu.cn

**Keywords:** Xinjiang, wheat, *Fusarium* mycotoxins, emerging mycotoxins

## Abstract

Wheat is a major staple crop in Xinjiang, China; however, comprehensive data on *Fusarium* mycotoxin contamination in wheat from this region remain limited. Despite recent observations of Fusarium head blight (FHB), few studies have characterized the mycotoxin profiles in wheat from Xinjiang, especially regarding emerging mycotoxins. This study aimed to systematically investigate the occurrence of both conventional and emerging mycotoxins in freshly harvested wheat from Xinjiang, to evaluate the effects of sampling year and geographical region on mycotoxin contamination levels, and to identify the *Fusarium* species responsible for mycotoxin production. A total of 151 freshly harvested wheat samples were collected from Southern and Northern Xinjiang in 2023 and 2024. Mycotoxins were quantified using high-performance liquid chromatography–tandem mass spectrometry (HPLC-MS/MS). *Fusarium* isolates were obtained and identified through the translation elongation factor 1-alpha (*TEF-1α*) gene sequencing. Genotyping was assessed by genotype-specific multiplex PCR, and mycotoxigenic potential was detected by rice culture assays. A high incidence (72.9%) of co-contamination with multiple mycotoxins was observed. Conventional mycotoxins such as deoxynivalenol (DON) and zearalenone (ZEN) were detected in 31.1% and 41.1% of samples. Notably, emerging mycotoxins, including enniatins (ENNs) and beauvericin (BEA), were present at significantly higher concentrations than those reported in some regions of China. Significant spatiotemporal variation was observed, with markedly higher contamination levels of emerging mycotoxins in 2024, particularly in Northern Xinjiang, where the symptoms of FHB epidemic occurred due to the humid climate and maize–wheat rotation system. *Fusarium graminearum* was identified as the primary producer of conventional mycotoxins, while *F. acuminatum* and *F. avenaceum* were mainly associated with emerging mycotoxins except BEA. This study provides the first comprehensive dataset on the co-occurrence of conventional and emerging *Fusarium* mycotoxins in wheat from Xinjiang and highlights significant spatiotemporal variations influenced by environmental factors. These findings underscore the necessity for continuous, region-specific monitoring and effective risk management strategies to address the evolving mycotoxin threat in Xinjiang’s wheat. Future research should focus on characterizing the populations of *Fusarium* toxin-producing fungi and the long-term impacts of mycotoxin exposure on food safety.

## 1. Introduction

Wheat (*Triticum aestivum* L.) is a crucial staple crop in Xinjiang, China, with a cultivation area that has remained at approximately 1.2 million hectares over the past five years. Notably, more than 96% of the wheat produced in this region is classified as high quality, a proportion substantially higher than that of other major wheat-producing areas in China [1]. However, in recent years, symptoms of Fusarium head blight (FHB) have been frequently observed from July to August, likely driven by changes in climate and agronomic practices, while limited information about FHB of wheat in Xinjiang has been reported.

Wheat is known to be contaminated with several conventional mycotoxins, including deoxynivalenol (DON), 3-acetyldeoxynivalenol (3ADON), 15-acetyldeoxynivalenol (15ADON), nivalenol (NIV), fusarenone X (4ANIV), and zearalenone (ZEN). Co-contamination of these mycotoxins has been documented in wheat from major wheat-producing regions of China [2]. To the best of our knowledge, only one survey has been conducted to assess the prevalence and concentration of DON in wheat harvested from Xinjiang Province in 2013, and DON was not detected in all 22 samples [3].

DON, 3ADON, 15ADON, NIV, and 4ANIV are mainly synthesized by the *Fusarium graminearum* species complex (FGSC). FGSC strains are categorized into three trichothecene chemotypes based on their dominant toxin profiles: the 3ADON chemotype (producing DON and primarily 3ADON), the 15ADON chemotype (producing DON and primarily 15ADON), and the NIV chemotype (producing NIV and its acetylated derivatives). The species composition and chemotype of FGSC are strongly influenced by geographical distribution. In China, the 15ADON chemotype of *F. graminearum* is predominant in the northern regions, the 3ADON chemotype of *F. asiaticum* dominates in the middle and lower reaches of the Yangtze River, and the 3ADON and NIV chemotypes of *F. asiaticum* and *F. meridionale* are both found in the northwest [2,4,5]. Although there has been no report about the species composition and chemotype of FGSC in wheat from Xinjiang, symptoms of Fusarium crown rot caused by *F. culmorum* have been observed in winter wheat in this region [6]. Advances in genomic sequencing technologies have greatly enhanced our understanding of the metabolic pathways of *Fusarium* species and allowed more accurate characterization of their mycotoxigenic potential in wheat [7].

Considerable attention has recently focused on a group of emerging mycotoxins, the in vivo toxicological and toxicokinetic data of which remain limited, such as beauvericin (BEA), enniatin B (ENNB), enniatin B_1_ (ENNB_1_), enniatin A (ENNA), and enniatin A_1_ (ENNA_1_). These compounds are cyclic depsipeptides produced by a variety of *Fusarium* species, such as *F. tricinctum*, *F. acuminatum*, *F. avenaceum*, and *F. torulosum* [8,9,10,11]. Recent research in China has shown that between 5.33% and 89.14% of wheat samples harvested from Anhui, Jiangsu, Hebei, Henan, Jiangsu, Shandong, Zhejiang, Shanxi, and Hubei Provinces are contaminated with at least one type of emerging mycotoxin, but the maximum concentration reported was less than 400 µg/kg [12,13,14,15,16].

Despite the increasing awareness of *Fusarium*-related risks in major wheat-producing regions of China, substantial knowledge gaps remain for Xinjiang. In particular, the natural occurrence and co-contamination patterns of both conventional (e.g., DON and ZEN) and emerging *Fusarium* mycotoxins (e.g., ENNs and BEA) have not been systematically investigated in this region. Furthermore, the composition of FGSC and distribution of trichothecene chemotypes in wheat from Xinjiang remain unknown. These gaps hinder the development of region-specific monitoring as well as limit the assessment of food safety risks associated with wheat production in this arid–semi-arid environment.

To address these deficiencies, this study integrates multi-mycotoxin analysis using high-performance liquid chromatography–tandem mass spectrometry (HPLC-MS/MS) with molecular identification of *Fusarium* isolates based on translation elongation factor 1-alpha (*TEF-1α*) sequencing. Genotype was assessed by specific multiplex PCR, and mycotoxigenic potential was detected by rice culture assays. This combined approach enables a comprehensive assessment of both the contamination landscape and the mycotoxigenic potential of the local *Fusarium* populations. Accordingly, the objectives of this study are to (1) quantify the natural occurrence of conventional and emerging *Fusarium* mycotoxins in freshly harvested wheat in Xinjiang; (2) evaluate the influence of sampling year and geographical region on contamination levels; and (3) identify the dominant *Fusarium* species and determine their mycotoxigenic potential. The results are expected to provide a scientific basis for informed mycotoxin control and risk management in Xinjiang’s wheat production chain.

## 2. Results and Discussion

### 2.1. Natural Occurrence of Fusarium Mycotoxins

The results of the analysis for the presence of *Fusarium* mycotoxins in wheat are shown in Table 1 and Appendix A. In total, 110 wheat samples (72.9%) out of 151 contained at least one type of *Fusarium* mycotoxin. ZEN and DON were the most prevalent, detected in 41.1% and 31.1% of wheat samples, followed by four emerging mycotoxins, ENNB (18.5%), BEA (14.6%), ENNB_1_ (8.61%), and ENNA_1_ (5.96%). The incidence of all other mycotoxins was below 5.00%. Notably, although ZEN and DON were frequently detected, their concentrations in all samples were below the maximum limits (MLs) established in Chinese standard GB 2761-2017 [17], as well as below the MLs in European Union (EU) regulations [18,19].

To the best of our knowledge, this is the first comprehensive dataset describing the co-occurrence of both conventional and emerging *Fusarium* mycotoxins in a large number of wheat samples from Xinjiang. Previous research in this region is extremely limited; only a study examining 22 samples from Northern Xinjiang for DON reported no contamination [3]. Co-contamination of DON, ZEN, BEA, and ENNs has been documented in major wheat-producing provinces, such as Zhejiang, Jiangsu, Anhui, Hubei, Henan, Shandong, Shanxi, and Hebei Provinces. Our findings indicate that the incidences of DON and ZEN in wheat in Xinjiang are lower than those reported in these regions over the past five years. However, the mean and maximum concentrations of ENNB, BEA, ENNB_1_, ENNA_1_, and ENNA in our positive samples were higher than those reported in these provinces. In contrast to the higher concentrations, the incidence of these emerging mycotoxins was lower than that observed in wheat and highland barley from Shandong and Xizang Provinces [12,20,21].

Historically, DON and ZEN have been the dominant *Fusarium* mycotoxins in Chinese wheat, with DON typically exhibiting higher prevalence and concentration than ZEN [2]. However, deviating from this national pattern, our study in Xinjiang observed a higher detection rate for ZEN (41.1%) than for DON (31.1%). Nevertheless, the mean concentration of DON (170 ± 23.1 μg/kg) still obviously exceeded that of ZEN (9.67 ± 1.04 μg/kg), aligning with trends observed in other regions [13,14]. Although the concentrations of DON and ZEN in all the samples are below the MLs in China, the considerable presence of both mycotoxins highlights the need for continuous monitoring in Xinjiang.

ENNs are major contaminants in cereals worldwide [22]. Global prevalence data from a meta-analysis of 6916 samples (2004–2024) rank Europe highest (56.6%), followed by Asia (24.6%), Africa (7.55%), and the Americas (1.76%). Importantly, mean concentrations of ENNB, ENNB_1_, ENNA_1_, and ENNA in African and European cereals are higher than those from Asia or America. The situation in Xinjiang presents a distinct profile: although the incidence is lower than in Asian, the mean concentration is substantially elevated [23]. These findings suggested that contamination of ENNs in Xinjiang showed a distinct pattern. Similarly, BEA is another emerging mycotoxin, reported in 9% to 100% of wheat or barley from numerous countries, and with a maximum level reaching 4870 μg/kg [14,16,24]. A consistent global trend was that BEA concentrations are generally lower than those of ENNs in wheat in Xinjiang, a pattern that was confirmed in the current investigation.

### 2.2. Effect of Sampling Timing and Regions on the Contents of Mycotoxins

To evaluate the influence of sampling year and region on mycotoxin contamination, we conducted a comparative analysis based on the total conventional (ZEN + DON + NIV + FB_1_ + 3ADON + 15ADON) and total emerging (BEA + ENNB + ENNB_1_ + ENNA + ENNA_1_ + DOM) mycotoxins. Although the mean level of total conventional mycotoxins in 2024 was substantially higher than that in 2023, the difference did not reach statistically significant levels. However, the concentration of total emerging mycotoxins in 2024 (Figure 1) was significantly higher than that in 2023 (*p* < 0.001). A significant geographical variation was also found, with wheat harvested from Northern Xinjiang (Figure 2) exhibiting significantly higher contamination levels of both conventional and emerging mycotoxins than Southern Xinjiang (*p* < 0.05).

The observed divergences in mycotoxin contamination across years and regions are likely driven by several interconnected factors, including temperature, humidity, crop rotation, and *Fusarium* species [25,26]. Although the mean temperatures in July and August are similar in Southern and Northern Xinjiang (approximately 25 °C), the mean cumulative precipitation in the north (300 mm) is substantially higher than in the south (60 mm) in recent years [27]. This climatic disparity is critical, as warmer and more humid conditions are known to be more conducive to *Fusarium* infestation and mycotoxin biosynthesis [25,28]. Extensive laboratory studies on members of FGSC, particularly *F. graminearum* and *F. meridionale*, have demonstrated that both fungal growth and the production of trichothecenes are highly dependent on water activity (a_w_) and temperature. While mycelial growth can occur at water activities as low as 0.90 a_w_, mycotoxin production requires considerably higher water activity, typically requiring a_w_ levels above 0.95 [29,30,31]. Crucially, the optimal conditions for toxin accumulation are often distinct from those for growth, with maximum DON production frequently reported at high water activity (0.995 a_w_) and temperatures around 25–30 °C [29,32]. Furthermore, even strains with a defined genetic background (e.g., 15ADON genotype) can exhibit dramatic shifts in their toxin production profiles (the relative abundance of DON, 3ADON, and 15ADON) in response to subtle changes in a_w_ and temperature [31]. The higher precipitation in Northern Xinjiang likely creates a microclimate on wheat spikes with sustained high levels of water activity that met or exceeded these critical thresholds during the grain-filling period. In contrast, the arid conditions in Southern Xinjiang would likely maintain the water activity around the wheat spike below the level that is adapted for substantial mycotoxin production, despite the conducive temperatures. This precipitation contrast may partly explain the higher mycotoxin levels observed in Northern Xinjiang.

Beyond environmental conditions, the composition of crop rotations is a critical driver of *Fusarium* infection, primarily through the pathogen carry-over via contaminated crop debris [26,33]. This study identifies a distinct regional divergence: Southern Xinjiang primarily employs cotton–wheat and orchard–wheat rotations, while Northern Xinjiang predominantly utilizes a wheat-silage corn system [1,34]. This rotational difference likely contributes to the observed contamination pattern. Extensive research has demonstrated that wheat following maize harbors significantly higher mycotoxin levels, a consequence of the abundant maize residues serving as a potent inoculum source [35,36]. These maize debris are ideal substrates for the pathogenic *Fusarium*, allowing the pathogen to survive saprophytically and produce abundant ascospores, which directly contribute to the primary inoculum for FHB of wheat. In contrast, the efficacy of cotton as a preceding crop in reducing DON accumulation in wheat, as reported by Selvaraj et al. [37], provides a mechanistic explanation for the lower contamination levels in Southern Xinjiang and aligns consistently with our findings. This can be attributed to the fact that cotton is a non-host crop for FGSC, thereby interrupting the disease cycle and drastically reducing the carryover of pathogenic inoculum. Furthermore, the drier field conditions and faster decomposition rates associated with cotton residues create an unfavorable microclimate for *Fusarium* survival and sporulation compared to the moist, residue-rich environment following maize harvest [35,37]. Thus, the substitution of maize with cotton in crop rotation systems acts as a cultural practice that directly diminishes the initial inoculum pressure, leading to a corresponding reduction in FHB incidence and DON contamination in wheat.

### 2.3. Species Composition and Chemotype of Fusarium Isolates

Thirty-eight *Fusarium* isolates were obtained and identified from wheat samples. *F. graminearum* (63.2%) was the predominantly detected species, followed by *F. avenaceum* (21.0%) and *F. acuminatum* (15.8%) (Figure 3). Genotype analysis revealed that 70.8% of the *F. graminearum* isolates were the 15ADON genotype (70.8%), while the remainders were the 3ADON genotype. Chemical analysis of rice cultures indicated distinct chemotype-specific profiles: the 15ADON genotype primarily produced DON, ZEN, and 15ADON, with minor amounts of 3ADON, while the 3ADON genotype produced DON, ZEN, and 3ADON, with trace levels of 15ADON. Production of ENNs was associated with both *F. acuminatum* and *F. avenaceum*.

The prevalence of the FGSC in Chinese wheat is well documented, with its species composition and chemotype distribution demonstrating strong geographical patterns. Previous studies have consistently identified the 15ADON chemotype of *F. graminearum* as the dominant population in the cool, semi-arid to semi-humid wheat-growing regions of Northern China [2,4,5]. Our findings from Xinjiang are consistent with this broader geographical trend, confirming that the 15ADON chemotype of *F. graminearum* is also the predominant contributor to FHB in this area. However, a notable divergence was observed in the distribution of the 3ADON chemotype. While it typically represents less than 10% of the population in other parts of Northern China [2,38,39,40], it accounted for 29.2% of the isolates in our collection. This discrepancy may be largely attributed to a marked north–south divergence within the region: the 3ADON chemotype was more frequent in Southern Xinjiang (37.5%), whereas the 15ADON producers were overwhelmingly dominant in Northern Xinjiang (75.0%). This clear regional differentiation underscores the influence of localized factors, such as microclimate and cropping systems, in shaping the population structure of *F. graminearum* [41,42].

Species within the *Fusarium tricinctum* species complex (FTSC), such as *F. arthrosporioides*, *F. acuminatum*, *F. avenaceum*, *F. tricinctum*, and *F. torulosum,* are established as key producers of ENNs in barley [8,9,11]. Our findings align with this (Table 2). We confirmed that ENNs could be produced by *F. acuminatum* and *F. avenaceum* in rice cultures. The structural diversity among the major ENNs (e.g., A, A_1_, B, B_1_), characterized by variations in their cyclic hexadepsipeptide core, is closely linked to their ionophoric properties and associated cytotoxicity, underscoring the critical relationship between metabolite structure and bioactivity [23]. Conversely, BEA was originally isolated from *Beauveria bassiana*, which is also known to be synthesized by various *Fusarium* complexes, including FTSC, *Fusarium fujikuroi* species complex (FFSC), and *Fusarium incarnatum*-*equiseti* species complex (FIESC) [10,11,43,44]. While 14.6% of wheat samples in Xinjiang were contaminated with BEA, our tested strains of *F. acuminatum* and *F. avenaceum* did not produce it. The absence of BEA production in our strains—despite its known occurrence in other FTSC members—highlights the metabolic diversity within the complex, whereby some strains produce BEA while others do not [45,46]. Previous molecular work shows that both BEA and ENNs are synthesized by the multifunctional enniatin/beauvericin synthetase (ESYN1), and that sequence/structural variation in ESYN1 correlates strongly with whether a given Fusarium strain produces BEA, ENNs, or both. Thus, the presence or absence of BEA biosynthetic capability in FTSC isolates likely reflects specific ESYN1 alleles and their structural differences. This pattern suggests our current strain set may be too limited to capture the full range of BEA-producing isolates and underscores the need for broader isolate sampling and, where possible, ESYN1 sequencing to link genotype to metabolite profile [46,47,48].

Our results, in conjunction with recent systematic reviews, underscore the broader relevance of *Fusarium* secondary metabolites as a source of bioactive natural compounds. Emerging mycotoxins such as the ENNs exhibit multifaceted and complex biological activities that extend well beyond the scope of traditional mycotoxins. As highlighted by the comprehensive systematic review of Coulet et al. (2025), ENNs display an intricate array of toxicological mechanisms in human cell models, ranging from ionophoric activity and mitochondrial dysfunction to oxidative stress, cell cycle perturbation, and apoptosis [23]. This diversity of bioactivity is intrinsically linked to their cyclic hexadepsipeptide chemical structure, underscoring a sophisticated interplay between metabolite configuration and cellular targets. Furthermore, the frequent co-occurrence of ENNs with other mycotoxins, most characteristically with the structurally related BEA. Consequently, categorizing ENNs merely as ‘food contaminants’ may understate their biological significance; they should be recognized as a class of natural products with complex pharmacological and toxicological potential. The comprehensive toxicological profiling of these emerging mycotoxins is therefore not only imperative for accurate safety assessment but also provides invaluable insights into the structure-activity relationships of cyclic depsipeptides and their potential relevance in broader biomedical contexts.

Although both conventional (e.g., DON, ZEN) and emerging (e.g., ENNs, BEA) *Fusarium* mycotoxins co-occurred in Xinjiang wheat, dietary exposure assessments in China have primarily focused on conventional mycotoxins. Most available biomonitoring evidence centers on DON, showing that its dietary intake can exceed internationally accepted safety thresholds in certain high-consumption populations. Biomonitoring evidence from China shows that dietary exposure to DON can exceed internationally accepted safety thresholds in certain high-consumption populations. For example, a human biomonitoring study in Henan Province reported a mean probable daily intake (PDI) of 1.61 μg/kg bw, exceeding the JECFA provisional maximum tolerable daily intake (PMTDI) of 1 μg/kg bw, with children and adolescents exhibiting the highest exposure levels [49]. In comparison, food-based exposure assessments from other regions of China (e.g., Zhejiang Province) generally report PDI values below the PMTDI but still identify subgroups with elevated exposure, reflecting substantial regional and dietary heterogeneity [50]. For ENNs and BEA, dietary exposure data in China remain scarce; however, a recent European Food Safety Authority (EFSA) assessment has concluded that, although acute health risks are unlikely, chronic exposure risks cannot be excluded due to limited toxicological and epidemiological evidence [51]. This uncertainty is particularly relevant in light of the comparatively high concentrations of ENNs and BEA observed in parts of wheat samples collected from Xinjiang.

Overall, both conventional and emerging *Fusarium* mycotoxins could be detected from wheat samples in Xinjiang. Notably, the mean and maximum concentrations of emerging mycotoxins in positive samples were higher than those reported in other regions in China. Regular monitoring for the presence of *Fusarium* mycotoxins is needed to verify the pattern of contamination distribution. Although conventional mycotoxins and ENNs are mainly produced by several species of FGSC and FTSC, BEA cannot be produced by these species. Large-scale fungal isolation and species-level identification are essential for understanding their distribution and toxigenic potential. In addition, unlike DON and ZEN, whose presence in food has been regulated by authorities, no limits have been set for ENNs and BEA in China [17]. There might be a concern with respect to chronic exposure, but no firm conclusion could be drawn, and a risk assessment was not possible for dietary exposure to ENNs, due to the overall lack of toxicity data. More occurrence data were needed for a future risk assessment [23].

## 3. Conclusions

This study reveals a high incidence (72.9%) of mycotoxin co-contamination in wheat from Xinjiang, with emerging mycotoxins (ENNs and BEA) reaching notably higher concentrations than those reported in Eastern China. Regional differences were observed, with Northern Xinjiang exhibiting higher contamination, which may be linked to the humid climate and maize–wheat rotation system. This study emphasizes the need for ongoing, region-specific mycotoxin monitoring and risk management strategies. Future research should focus on expanding the analysis of *Fusarium* species and evaluating the long-term impact of mycotoxin exposure on food safety.

## 4. Materials and Methods

### 4.1. Chemicals and Reagents

A multi-toxin standard solution (Catalog number: DZ-STD) containing reference standards for *Fusarium* toxins, including DON, 3ADON, 15ADON, NIV, 4ANIV, ZEN, ENNB, ENNB_1_, ENNA, ENNA_1_, BEA, FB_1_, fumonisin B_2_ (FB_2_), fumonisin B_3_ (FB_3_), Diacetoxyscirpenol (DAS), T-2 toxin (T-2), HT-2 toxin (HT-2), Neosolaniol (NEO), and DOM was commissioned to Pribolab (Qindao, China) for development. Liquid chromatography (LC)-grade solvents (acetonitrile and methanol) were purchased from Merck KGaA (Darmstadt, Germany). Deionized water (resistivity < 8 MΩ/cm) was prepared using a Milli-Q water purification system (Millipore, Bedford, MA, USA). Each solvent was passed through cellulose filters with a 0.22 μm pore diameter (Jinteng Experimental Equipment, Tianjin, China) before use.

### 4.2. Sample Collection

A total of 151 freshly harvested wheat samples were randomly collected from two regions (Region 1: southern, Region 2: northern) in Xinjiang according to the planting areas (Figure 4). The number of wheat samples collected from Southern Xinjiang was 50 and 57, and Northern Xinjiang was 23 and 21 in 2023 and 2024, respectively. The sampling plan involved 10 incremental samples, resulting in an aggregate sample of 2.0 kg [52]. The grains were obtained after post-harvest cleaning and drying stages (up to a maximum of 55 °C) in the storage unit. Each sample was packed in a paper bag and stored at −4 °C for immediate mycotoxin analysis.

### 4.3. Mycotoxin Analysis and Method Validation

Each 2.0 kg wheat sample was thoroughly homogenized using the quartering method. A 500 g subsample was weighed, milled, and passed completely through a 20-mesh sieve. A quick, easy, cheap, effective, rugged, and safe (QuEChERS) method was used for sample preparation, and all the *Fusarium* mycotoxins were investigated for each sample by HPLC-MS/MS [21]. The detectability of the *Fusarium* mycotoxins was quantified by the limits of detection (LODs) and the limits of quantification (LOQs), which were estimated based on signal-to-noise ratios of 3/1 and 10/1, respectively. For quantification, a matrix-matched calibration curve comprising a blank and six spiked levels was built (Appendix A). The coefficients of correlation (*R*^2^) for the calibration curve ranged from 0.997 to 0.999. Three different concentrations of each *Fusarium* mycotoxin were spiked in triplicate into the blank matrix. After sample preparation and HPLC-MS/MS measurement, the recovery was calculated. The recoveries obtained for all 19 toxins ranged from 76.8% to 113%, with associated relative standard deviations (RSDs) all below 8.7% (Table 3).

### 4.4. Fungi Isolation

Fungi were isolated from all the wheat samples. Wheat seeds from each sample were soaked in 1% sodium hypochlorite for 2 min, rinsed in sterile water for 2 min, placed onto PDA supplemented with tetracycline sulfate, and incubated at 25 °C for 5 days. *Fusarium* clonies were transferred to liquid mung bean medium and incubated under fluorescent cool-white light at 25 °C [53]. All *Fusarium* isolates were purified by subculturing single conidia, and conidial suspensions were stored in 30% glycerol at −80 °C [10].

### 4.5. Species Composition of Fusarium Isolates

Genomic DNA was extracted using the UE Genomic DNA Mini Preparation Kit (UElandy Biotechnology Co., Ltd., Suzhou, China). A partial sequence of the translation *TEF-1α* gene was amplified by PCR following previously described primers and protocols [28]. PCR products were sequenced by Shanghai Shenggong Biotechnological Ltd. (Shanghai, China). The sequences were assembled and aligned using DNASTAR Lasergene v7.1 software (Madison, WI, USA). Reference *TEF-1α* sequences representing known *Fusarium* species were downloaded from GenBank. Phylogenetic relationships were inferred using the Neighbor-Joining method in MEGA 7.0 (University Park, PA, USA), with a bootstrap value of 1000 replicates and a cutoff of 50% [54].

### 4.6. Genotype and Mycotoxin Production of Fusarium Isolates

Trichothecene genotype-specific multiplex PCR assays based on the *Tri11* gene were conducted as described in previous studies [55]. The *Tri11* gene primers produced amplified target DNA fragments of approximately 297 bp, 334 bp, and 497 bp, corresponding to the 15ADON, 3ADON, and NIV genotypes, respectively.

Mycotoxin production was assessed using rice cultures. Briefly, 50 g of long-grain rice and 15 mL of sterile deionized water were added to 250 mL glass flasks, and this medium was autoclaved twice on alternating days before being inoculated with 5 mycelium plugs (0.5 cm diameter) taken from 3-day-old cultures of each strain. Flasks were incubated at 25 °C for 21 days, and developed cultures were ground and stored at −20 °C until use. Non-inoculated rice cultures were used as a negative control. All *Fusarium* mycotoxins were quantified by HPLC-MS/MS [20].

### 4.7. Statistical Analysis

The experimental data were expressed as means ± standard errors of the mean and are accompanied by the number of observations (*n*). All data were analyzed using IBM SPSS Statistics 25 software (Winchester, UK). The Kolmogorov–Smirnov test was used to compare the levels of conventional and emerging *Fusarium* mycotoxins in various sampling years and regions. Data visualization and graph generation were performed using GraphPad Prism version 10.1.2 (Boston, MA, USA).

## Figures and Tables

**Figure 1 toxins-17-00591-f001:**
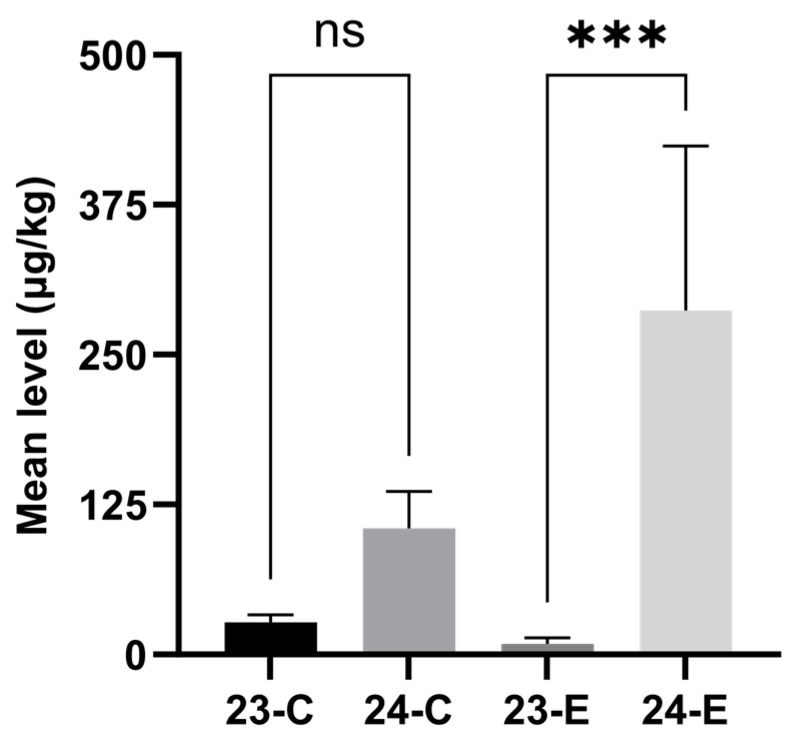
Comparison of the mean levels of total conventional and emerging *Fusarium* mycotoxins in wheat from Xinjiang between 2023 and 2024. 23-C and 24-C: Mean level of total conventional mycotoxins in Xinjiang in 2023 and 2024; 23-E and 24-E: Mean level of total emerging mycotoxins in Xinjiang in 2023 and 2024. *** Significant at *p* < 0.001, ns significant at *p* > 0.05.

**Figure 2 toxins-17-00591-f002:**
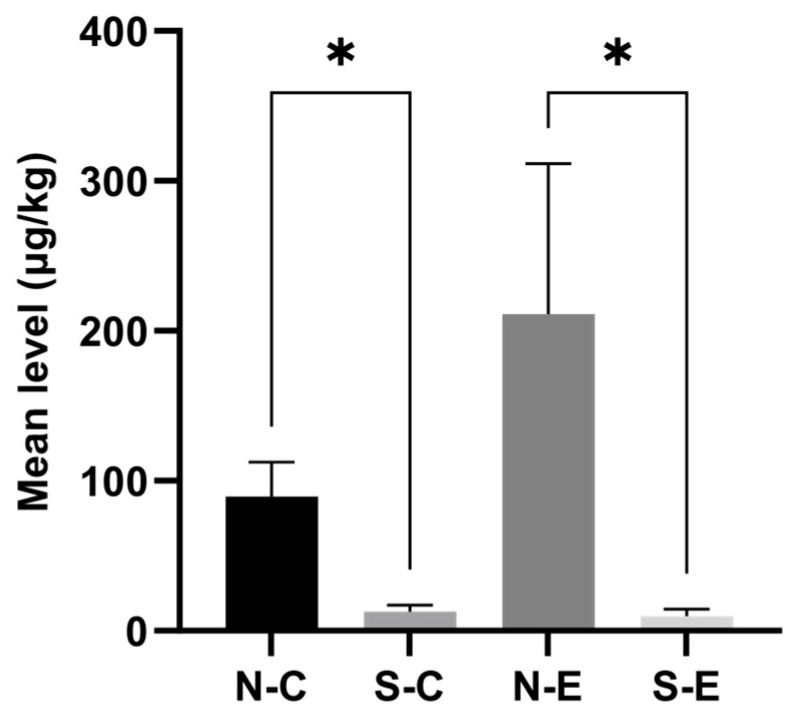
Comparison of the mean levels of total conventional or emerging *Fusarium* mycotoxins in wheat from Northern and Southern Xinjiang. N-C and S-C: Mean levels of total conventional mycotoxins in Northern and Southern Xinjiang; N-E and S-E: Mean levels of total emerging mycotoxins in Northern and Southern Xinjiang. * Significant at *p* = 0.05.

**Figure 3 toxins-17-00591-f003:**
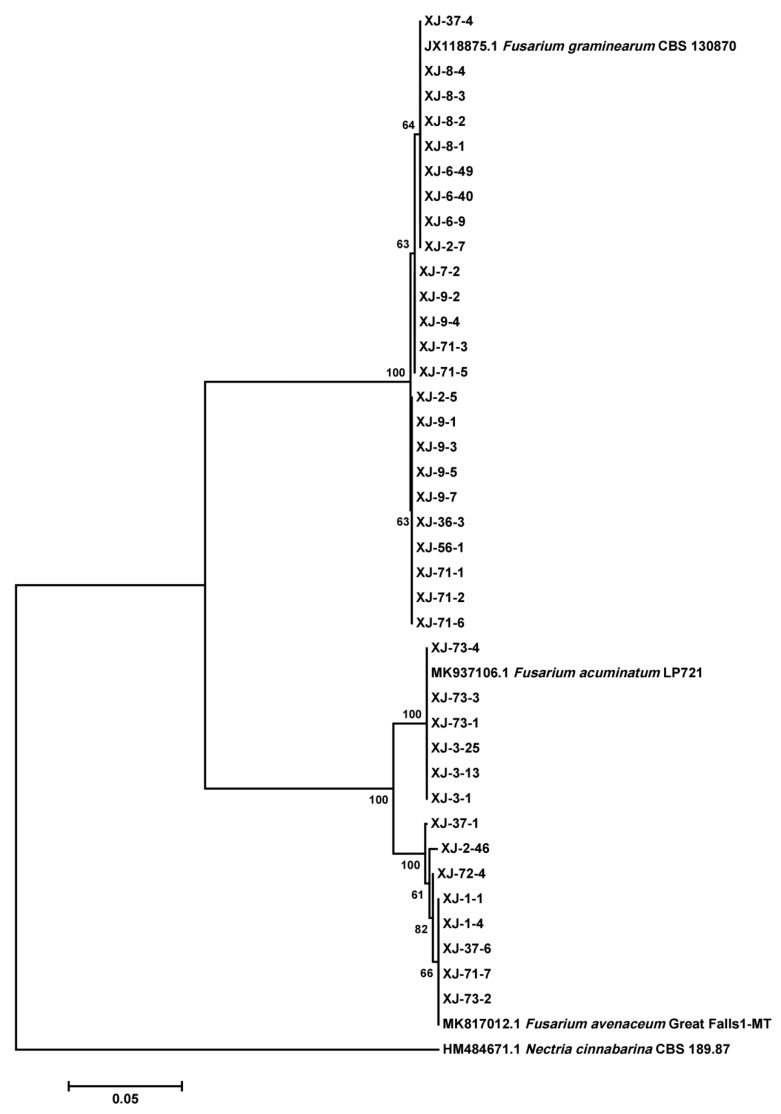
Phylogenetic tree of *Fusarium* isolates based on partial *TEF-1α* gene sequences.

**Figure 4 toxins-17-00591-f004:**
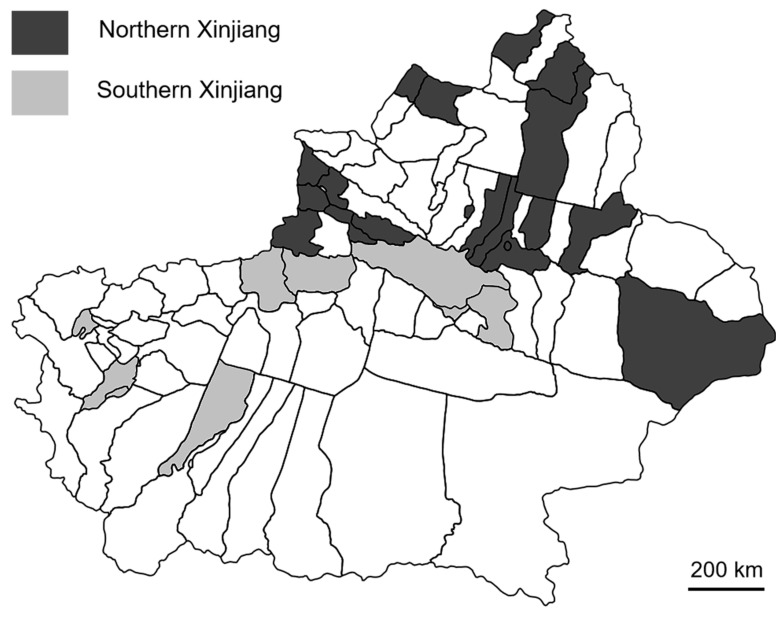
Map of Xinjiang showing its two geographical regions for sample collection.

**Table 1 toxins-17-00591-t001:** Profile of *Fusarium* toxins contamination in freshly harvested wheat samples from Xinjiang.

Mycotoxin	Positive Samples (*n*) ^a^	Incidence of Contamination (%)	Mean of Positive Samples (μg/kg)	Contamination Range (μg/kg)	ML (μg/kg)	
					China	EU
ZEN	62	41.1	9.67 ± 1.04	0.96–45.5	60	100
DON	47	31.1	170 ± 23.1	10.3–831	1000	1500
ENNB	28	18.5	319 ± 62.8	5.60–3305	/	/
BEA	22	14.6	115 ± 16.1	6.35–829	/	/
ENNB_1_	13	8.61	565 ± 97.2	7.03–3832	/	/
ENNA_1_	9	5.96	341 ± 51.0	12.5–1651	/	/
3ADON	7	4.64	130 ± 12.5	27.2–257	/	/
15ADON	7	4.64	76.2 ± 7.63	16.0–188	/	/
DOM ^b^	6	3.97	81.5 ± 7.22	32.0–135	/	/
ENNA	6	3.97	110 ± 14.4	6.02–393	/	/
NIV	3	1.99	33.7 ± 4.11	6.42–86.9	/	/
FB_1_ ^c^	1	0.66	14.4	14.4	/	/

^a^ The contamination concentrations of positive samples ≥ LOQ. ^b^ DOM: Deepoxy-deoxynivalenol. ^c^ FB_1_: Fumonisin B_1_.

**Table 2 toxins-17-00591-t002:** *Fusarium toxins* production profiles of *Fusarium* isolates in rice culture assays.

Mycotoxin	Mean Level of *Fusarium* toxins (μg/g)			
	*F. graminearum*-15ADON	*F. graminearum*-3ADON	*F. acuminatum*	*F. avenaceum*
DON	232 ± 34.6	258 ± 45.9	-	-
3ADON	9.46 ± 1.69	73.1 ± 42.6	-	-
15ADON	41.7 ± 5.74	34.2 ± 5.47	-	-
ZEN	124 ± 23.0	176 ± 62.6	-	-
ENNA	-	-	29.1 ± 7.99	0.75 ± 0.51
ENNA_1_	-	-	150 ± 31.2	13.1 ± 6.87
ENNB	-	-	234 ± 32.3	123 ± 55.2
ENNB_1_	-	-	267 ± 50.4	57.0 ± 27.2

**Table 3 toxins-17-00591-t003:** The linearity, correlation coefficient, LODs, LOQs, spiked, recoveries and RSDs values of 19 *Fusarium* mycotoxins.

Mycotoxin	Linearity (μg/kg)	*R* ^2^	LOD (μg/kg)	LOQ (μg/kg)	Spiked (μg/kg)	Recovery (%)	RSD (%)
DON	10.0~2000	0.999	5.0	10.0	10.0, 20.0, 100.0	87.6, 96.8, 95.2	6.8, 5.9, 1.8
3ADON	10.0~2000	0.999	5.0	10.0	10.0, 20.0, 100.0	86.4, 108, 104	4.3, 3.6, 3.2
15ADON	10.0~2000	0.998	5.0	10.0	10.0, 20.0, 100.0	84.4, 103, 106	3.6, 3.0, 2.9
4ANIV	10.0~2000	0.999	3.0	10.0	5.0, 10.0, 50.0	84.2, 101, 94.7	6.9, 5.8, 4.7
NIV	5.0~1000	0.999	2.0	5.0	5.0, 10.0, 50.0	83.3, 88.6, 90.7	2.8, 3.1, 2.6
FB_1_	5.0~1000	0.999	2.0	5.0	5.0, 10.0, 50.0	85.8, 84.9, 83.4	2.6, 1.5, 1.7
FB_2_	5.0~1000	0.998	2.0	5.0	5.0, 10.0, 50.0	84.7, 88.3, 85.9	6.8, 3.2, 6.9
FB_3_	5.0~1000	0.998	2.0	5.0	5.0, 10.0, 50.0	96.1, 92.9, 95.7	6.5, 6.1, 8.2
T-2	5.0~1000	0.998	2.0	5.0	5.0, 10.0, 50.0	92.2, 105, 97.5	8.6, 3.2, 3.2
HT-2	5.0~1000	0.997	2.0	5.0	5.0, 10.0, 50.0	102, 109, 113	7.5, 2.4, 3.1
NEO	5.0~1000	0.998	2.0	5.0	5.0, 10.0, 50.0	89.6, 95.3, 93.5	8.7, 2.9, 4.9
DOM	5.0~1000	0.997	2.0	5.0	5.0, 10.0, 50.0	109, 102, 96.4	3.6, 5.9, 3.6
DAS	5.0~1000	0.998	2.0	5.0	5.0, 10.0, 50.0	94.1, 108, 95.2	5.3, 5.8, 6.2
BEA	5.0~1000	0.999	2.0	5.0	5.0, 10.0, 50.0	83.2, 87.3, 86.0	3.2, 3.2, 2.3
ENNA	5.0~1000	0.999	2.0	5.0	5.0, 10.0, 50.0	76.8, 84.3, 78.8	1.3, 2.5, 0.8
ENNA_1_	5.0~1000	0.999	2.0	5.0	5.0, 10.0, 50.0	86.5, 85.2, 78.9	2.2, 2.7, 2.4
ENNB	5.0~1000	0.998	2.0	5.0	5.0, 10.0, 50.0	79.9, 85.6, 89.2	1.8, 2.4, 1.3
ENNB_1_	5.0~1000	0.999	2.0	5.0	5.0, 10.0, 50.0	86.8, 90.0, 86.4	2.3, 2.9, 1.9
ZEN	1.0~200	0.999	0.3	1.0	1.0, 2.0, 10.0	76.8, 84.3, 78.8	2.9, 2.1, 1.6

## Data Availability

The data that support the findings of this study are available from the corresponding author, upon reasonable request due to the concerns from some local authorities and wheat producers in Xinjiang.

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
