# Peer review of "Natural Occurrence of Conventional and Emerging *Fusarium* Mycotoxins in Freshly Harvested Wheat Samples in Xinjiang, China"

_toxins, 2025, doi:10.3390/toxins17120591_

Round 1
Reviewer 1 Report
Comments and Suggestions for Authors
The paper: „Natural Occurrence of Conventional and Emerging Fusarium Mycotoxins in the Freshly Harvested Wheat Samples in Xinjiang, China„ is of interest from an applied point of view as well as in terms of novelty. The paper aimed to investigate contamination by both conventional and emerging Fusarium toxins in wheat samples collected from different regions of China. I consider that the paper is well-documented, the results are scientifically validated, and I appreciate the fact that elements of method validation such as LOQ and LOD were presented, which lends credibility to the study. Additionally, I appreciate the significant number of samples analyzed (151) and the mycotoxins studied (12).
I consider only a few additions are necessary:
- In Table 1, please insert a column with the number of samples with values above the maximum admited limit (MAL) and interpret the results in relation to these.
- Also, insert a table with the MAL for all analyzed mycotoxins, the values allowed in China, and compare them with other regulations (EU, USA, etc.).
- In the methods section, please detail the procedure for determining mycotoxins using HPLC-MS/MS (equipment, mobile phases, program, m/Z values), possibly in the form of a table.
- In the supplementary material or methodology, insert a chromatogram/MS spectrum regarding the analyzed standards and a sample model.
Author Response
Comment 1: In Table 1, please insert a column with the number of samples with values above the maximum limit (ML) and interpret the results in relation to these.
Response 1: Thank you for the reviewer's valuable comment. But no samples we tested showed values exceeding the ML threshold. This is described in the main text.
Comment 2: Also, insert a table with the MAL for all analyzed mycotoxins, the values allowed in China, and compare them with other regulations (EU, USA, etc.).
Response 2: Thank you for the reviewer's valuable comment. We have inserted two columns in Table 1 containing the ML values of all analyzed mycotoxins in wheat according to Chinese and EU regulations.
Comment 3: In the methods section, please detail the procedure for determining mycotoxins using HPLC-MS/MS (equipment, mobile phases, program, m/Z values), possibly in the form of a table.
Response 3: Thank you for the valuable comments from the reviewers. Our methods for mycotoxin extraction and HPLC-MS/MS detection are based on previous research in our laboratory. The methods section of Zhang et al. (2023) already provides a detailed table of the steps (equipment, mobile phases, program, m/Z values, etc.) for HPLC-MS/MS determination of mycotoxins; therefore, this study will not repeat them in detail to reduce information duplication (https://doi.org/10.1007/s12550-023-00487-1).
Comment 4: In the supplementary material or methodology, insert a chromatogram/MS spectrum regarding the analyzed standards and a sample model.
Response 4: We appreciate the reviewer's valuable comment. Added.

Reviewer 2 Report
Comments and Suggestions for Authors
Dear Authors
I have carefully read the manuscript (toxins-3993991). This manuscript presents valuable baseline data on the natural occurrence of conventional and emerging Fusarium mycotoxins in freshly harvested wheat from Xinjiang, China. The study integrates field sampling, multi-toxin LC-MS/MS analysis, Fusarium isolation, molecular genotyping, and in vitro toxin profiling. While the work is timely and regionally important. However, some minor concerns must be addressed to enhance the scientific clarity and presentation of the manuscript. Below are section-wise detailed comments.
General/Major comments
- The manuscript contains numerous grammatical, punctuation, spacing, and spelling errors. It is recommended that the manuscript be revised thoroughly by a native English speaker or a professional language editing service.
- The manuscript lacks sufficient updated references to support key claims, most important mechanisms, and comparisons with previous studies. I strongly recommend that the authors include the recommended references at appropriate points to enhance the novelty of the work.
Comments
Abstract
- The wording in the abstract is imprecise and weakens the scientific impact of the study. A complete revision is necessary to improve the clarity and scientific tone.
- Please follow the abstract structure 1. Background, 2. Aim, 3. Methodology, 4. Results and 5. Conclusion.
- Line 10-11: “To comprehensively investigate…” is grammatically incomplete; revise to a full sentence.
- Include a concise summary of the main findings and potential future directions at the end of the abstract.
Introduction
- Line 41-43: Insert Yang et al. (2021) when discussing climate- or stress-induced physiological changes in crops, emphasizing that plant genetic regulators can alter host susceptibility to pathogens and mycotoxin accumulation (https://doi.org/10.1111/tpj.15285).
- Line 64-72: Cite Lodi et al. (2025) to highlight that advances in whole-genome sequencing of fungi are rapidly improving understanding of fungal metabolic pathways, supporting the importance of characterizing Fusarium toxigenic potential in Xinjiang wheat (doi: 10.1038/s41597-025-05798-9).
- Line 80-82: Remove subjective wording like “anticipated”; use objective scientific tone.
- Consider elaborating more on the research background, clearly stating the knowledge gap, the problem being addressed, and how the present study aims to provide the solution in the introduction section.
- Conclude the introduction with a clear hypothesis and a concise study aim statement.
Results & Discussion
- Line 147-154: Climatic factors require quantification; discuss specific humidity thresholds required for Fusarium growth.
- Line 155-165: More detail is needed regarding how cotton rotations reduce inoculum compared to maize rotations.
- Line 194-202: Incorporate Gong at al. (2024) when discussing ENN production by Fusarium species, may help highlight the need to link structural features of metabolites to biological activity, an area still underdeveloped for ENNs in cereals (doi: https://doi.org/10.1016/j.ijbiomac.2024.137232).
- Line 194-202: Clarify why BEA was detected in field samples but not in vitro; discuss substrate-dependent biosynthesis.
- Line 203: Discussion lacks a section on public health implications; add risk relevance based on Chinese dietary exposure.
- Line 203-215: Include Xiao et al. (2025) to highlight the broader relevance of bioactive natural compounds and to argue that emerging Fusarium mycotoxins like ENNs may possess complex bioactivities requiring toxicological elucidation (https://doi.org/10.1016/j.ijbiomac.2024.139095).
- Consider integrating a conceptual model summarizing environment, Fusarium population and toxin outcomes.
Conclusion
- Conclusion section should be in 3-4 concise sentences emphasizing key findings, significance, limitations, and future prospects.
Materials & Methods
- The Materials and Methods section is well-described and ensures reproducibility.
- Line 217-226: Provide catalog numbers and suppliers for all standards to ensure reproducibility.
- Line 231-234: Comment: Clarify whether drying to 55 °C may degrade heat-sensitive toxins such as ENNs.
Author Response
Comment 1: The wording in the abstract is imprecise and weakens the scientific impact of the study. A complete revision is necessary to improve the clarity and scientific tone.
Response 1: We appreciate the reviewer's valuable comment and have revised the abstract to be more rigorous, clearer, and more scientifically sound.
Comment 2: Please follow the abstract structure 1. Background, 2. Aim, 3. Methodology, 4. Results and 5. Conclusion.
Response 2: We appreciate the reviewer's valuable comment and have written the abstract according to the required structure.
Comment 3: Line 10-11: “To comprehensively investigate…” is grammatically incomplete; revise to a full sentence.
Response 3: We appreciate the reviewer's valuable comment and agree with it. We have revised "To comprehensively investigate…" to "This study aimed to comprehensively investigate…".
Comment 4: Include a concise summary of the main findings and potential future directions at the end of the abstract.
Response 4: We appreciate the reviewer's valuable comment and have added a brief summary at the end of the abstract.
Comment 5 : Line 41-43: Insert Yang et al. (2021) when discussing climate- or stress-induced physiological changes in crops, emphasizing that plant genetic regulators can alter host susceptibility to pathogens and mycotoxin accumulation (https://doi.org/10.1111/tpj.15285).
Response 5: We thank the reviewer for this comment. But, this reference name was “OsTTG1, a WD40 repeat gene, regulates anthocyanin biosynthesis in rice”, which was not applicable to our study. Furthermore, this study did not focus on the host plant's genetic regulatory factors' susceptibility to pathogens and the accumulation of fungal toxins.
Comment 6: Line 64-72: Cite Lodi et al. (2025) to highlight that advances in whole-genome sequencing of fungi are rapidly improving understanding of fungal metabolic pathways, supporting the importance of characterizing Fusarium toxigenic potential in Xinjiang wheat (doi: 10.1038/s41597-025-05798-9).
Response 6: We appreciate the reviewer's valuable comment and agree that "advances in whole-genome sequencing of fungi are rapidly improving understanding of fungal metabolic pathways, supporting the importance of characterizing Fusarium toxigenic potential in Xinjiang wheat" However, the study by Lodi et al. (2025) (doi: 10.1038/s41597-025-05798-9) is too weakly relevant to our study. Therefore, we cite the study by Walker et al. (2025) (https://doi.org/10.3390/toxins17060284) to emphasize the contribution of genome research to understanding fungal metabolic pathways and support the characterization of the toxin-producing potential of Fusarium in Xinjiang wheat.
Comment 7: Line 80-82: Remove subjective wording like “anticipated”; use objective scientific tone.
Response 7: We appreciate the reviewer's valuable comment. We have removed "It is anticipated that the results will…" and changed it to "The results are expected to…".
Comment 8: Consider elaborating more on the research background, clearly stating the knowledge gap, the problem being addressed, and how the present study aims to provide the solution in the introduction section.
Response 8: We appreciate the reviewer's valuable comment and agree with their suggestion to elaborate more on the research background in the introduction, more clearly explaining the knowledge gaps, the problems this study aims to solve, and how it intends to provide solutions.
Comment 9: Conclude the introduction with a clear hypothesis and a concise study aim statement.
Response 9: We appreciate the reviewer's valuable comment and have provided some clear hypotheses and concise research objectives at the end of the introduction.
Results & Discussion
Comment 10: Line 147-154: Climatic factors require quantification; discuss specific humidity thresholds required for Fusarium growth.
Response 10: We appreciate the reviewer's valuable comment. While we lack field meteorological data to quantify climatic factors, we have cited some literature to quantify and discuss the specific humidity (water activity) thresholds required for Fusarium growth and mycotoxin production.
Comment 11: Line 155-165: More detail is needed regarding how cotton rotations reduce inoculum compared to maize rotations.
Response 11: We appreciate the reviewer's valuable comment, and we have discussed in more detail how cotton rotation reduces pathogens and mycotoxins compared to maize rotation.
Comment 12: Line 194-202: Incorporate Gong at al. (2024) when discussing ENN production by Fusarium species, may help highlight the need to link structural features of metabolites to biological activity, an area still underdeveloped for ENNs in cereals (doi: https://doi.org/10.1016/j.ijbiomac.2024.137232).
Response 12: We appreciate the reviewer's valuable comment, but the research paper by Gong at al. (2024), titled "Separation, purification, structure characterization, and immune activity of a polysaccharide from Alocasia cucullata obtained by freeze-thaw treatment," is less relevant to our study. We used a more relevant article (https://doi.org/10.1111/1541-4337.70270) to emphasize the necessity of linking the structural characteristics of metabolites to their biological activities.
Comment 13: Line 194-202: Clarify why BEA was detected in field samples but not in vitro; discuss substrate-dependent biosynthesis.
Response 13: We appreciate the reviewer's valuable comment. Although we have already clarified these points in the paper, we have supplemented our analysis with further analysis on the relationship between intraspecific metabolic diversity, chemotype, and the origin of biosynthesis, thus giving the phenomenological analysis a greater theoretical depth.
Comment 14: Line 203: Discussion lacks a section on public health implications; add risk relevance based on Chinese dietary exposure.
Response 14: We appreciate the reviewer's valuable comment and have added a discussion on the public health impacts of Fusarium toxins.
Comment 15: Line 203-215: Include Xiao et al. (2025) to highlight the broader relevance of bioactive natural compounds and to argue that emerging Fusarium mycotoxins like ENNs may possess complex bioactivities requiring toxicological elucidation (https://doi.org/10.1016/j.ijbiomac.2024.139095).
Response 15: We appreciate the reviewer's valuable comment. We agree with this comment, but the research by Xiao et al. (2025) is not relevant to the topic of this study, so we cite the review by Coulet et al. (2025) to support this point. (https://doi.org/10.1111/1541-4337.70270).
Comment 16: Consider integrating a conceptual model summarizing environment, Fusarium population and toxin outcomes.
Response 16: Thanks for the reviewer’s comment. As we know, more environmental data, population dynamics data, and mycotoxin contamination data are required to develop a robust model. We will subsequently intensify our research efforts in these areas, aiming to establish a suitable model for assessing the mechanisms behind toxin contamination in Xinjiang wheat.
Comment 17: Conclusion section should be in 3-4 concise sentences emphasizing key findings, significance, limitations, and future prospects.
Response 17: We appreciate the reviewer's valuable comment. We have simplified the conclusion.
Comment 18: Line 217-226: Provide catalog numbers and suppliers for all standards to ensure reproducibility.
Response 18: We appreciate the reviewer's valuable comment and have added the catalog number and supplier information for the muti-toxin standard solution.
Comment 19: Line 231-234: Comment: Clarify whether drying to 55 °C may degrade heat-sensitive toxins such as ENNs.
Response 19: We appreciate the reviewer's valuable comment. The review by France Coulet et al. (2025) emphasizes that thermal degradation depends on multiple factors: matrix type (whole grain, flour, dough), heating time, humidity, etc( https://doi.org/10.1111/1541-4337.70270). EFSA states in its scientific opinion that ENNs are "substantially stable" during grain processing, including drying and storage(https://doi.org/10.2903/j.efsa.2014.3802). At a relatively mild temperature of 55°C, even if heated for a considerable period of time (e.g., several hours or even days), the degradation of ENNs in wheat is likely to be very limited, slow, or even negligible.

Round 2
Reviewer 1 Report
Comments and Suggestions for Authors
The authors took into account the reviewers' suggestions and made all the requested changes. I would have only one comment: in the supplementary material, the chromatograms should indicate what each represents.
Author Response
Comment: In the supplementary material, the chromatograms should indicate what each represents.
Response: Thank you for the reviewer's valuable comment. We have included annotations regarding the chromatogram content in the supplementary material file.
